# High-Value Patents Recognition with Random Forest and Enhanced Fire Hawk Optimization Algorithm

**DOI:** 10.3390/biomimetics10090561

**Published:** 2025-08-23

**Authors:** Xiaona Yao, Huijia Li, Sili Wang

**Affiliations:** 1Key Laboratory of Ecological Safety and Sustainable Development in Arid Lands, Northwest Institute of Eco-Environment and Resources, Chinese Academy of Sciences, Lanzhou 730000, China; lihj@llas.ac.cn (H.L.); wangsl@llas.ac.cn (S.W.); 2Key Laboratory of Knowledge Computing and Intelligent Decision, Lanzhou 730000, China

**Keywords:** high-value patents recognition, FHO, inertial weight, levy flight, t-distribution perturbation, random forest

## Abstract

High-value patents are a key indicator of new product development, the emergence of innovative technology, and a source of innovation incentives. Multiple studies have shown that patent value exhibits a significantly skewed distribution, with only about 10% of patents having high value. Identifying high-value patents from a large volume of patent data in advance has become a crucial problem that needs to be addressed urgently. However, current machine learning methods often rely on manual hyperparameter tuning, which is time-consuming and prone to suboptimal results. Existing optimization algorithms also suffer from slow convergence and local optima issues, limiting their effectiveness on complex patent datasets. In this paper, machine learning and intelligent optimization algorithms are combined to process and analyze the patent data. The Fire Hawk Optimization Algorithm (FHO) is a novel intelligence algorithm suggested in recent years, inspired by the process in nature where Fire Hawks capture prey by setting fires. This paper firstly proposes the Enhanced Fire Hawk Optimizer (EFHO), which combines four strategies, namely adaptive tent chaotic mapping, hunting prey, adding the inertial weight, and enhanced flee strategy to address the weakness of FHO development. Benchmark tests demonstrate EFHO’s superior convergence speed, accuracy, and robustness across standard optimization benchmarks. As a representative real-world application, EFHO is employed to optimize Random Forest hyperparameters for high-value patent recognition. While other intelligent optimizers could be applied, EFHO effectively overcomes common issues like slow convergence and local optima trapping. Compared to other classification methods, the EFHO-optimized Random Forest achieves superior accuracy and classification stability. This study fills a research gap in effective hyperparameter tuning for patent recognition and demonstrates EFHO’s practical value on real-world patent datasets.

## 1. Introduction

High-value patents recognition, or patent evaluation, or patent value assessment, refers to the process of identifying patents that hold high technical, market, economic, legal or strategic value [1]. High-value patents are a key indicator of new product development, the emergence of innovative technology, and a source of innovation incentives [2]. These patents carry important innovations and technological breakthroughs, and have a significant influence on the development of business and society [3]. From an enterprise perspective, the identification and acquisition of high-value patents not only guide technological development and optimize resource allocation but also prevent redundant research efforts and decrease risks associated with innovation. Moreover, such patents enhance technological barriers and protect core competitive advantages, thereby promoting sustainable growth and long-term profitability. At the national level, and for science and technology management departments, accurately identifying high-value patents supports the efficient allocation of resources and informs the development of effective policy strategies [4]. However, low-value patents consume a lot of resources for innovation, which may cause firms to innovate more slowly if their low-value patents are not converted or commercialized. This would impede a nation’s industrial development, technological advancement, productivity growth, and economic prosperity [5,6]. Therefore, the high-value patents recognition from a huge number of patents is crucial for the cultivation, application, protection and management of high-value patents in government departments, scientific research institutions and enterprises, and is also a hot topic of academic research [7,8].

With the progress of complex networks and machine learning methods, it is possible to identify high-value patents from large-scale data [9]. Machine learning can effectively deal with large-scale, high-dimensional and structured patent data, extract deep-seated features of patents, and improve the accuracy of patent value identification [10,11,12]. Hido et al. [13] introduce a machine learning and text mining-based tool that assesses patent application quality by computing a score, which predicts the likelihood of approval, and outperforms conventional methods by utilizing a large dataset and a new statistical prediction model. Wu et al. [14] established a patent quality classification model using support vector machine, self-organizing map and kernel principal component analysis, and developed a patent quality analysis and classification system based on it, and the experimental results showed the advantages of the method. Based on multiple relevant metrics that can be accessed immediately after patent issuance, Lee et al. [15] provide a machine learning method for early detection of emergent inventions. Their method can help with responsive technology forecasting and planning, as demonstrated by the pharmaceutical technology case. A deep learning analytical approach was also used by Trappey et al. [16] to automate the evaluation of patent value in the Internet of Things domain. Using the provided dataset, they first identified important patent value indicators using principal component analysis. A Deep Neural Network (DNN) was utilized to forecast values after a total of 11 value indicators were chosen. The outcomes demonstrated that the DNN model outperformed the conventional Back-Propagation Neural Network technology in terms of accuracy. Kwon et al. [17] address the identification of promising inventions by using patent-based machine learning techniques that incorporate the quality of knowledge accumulation as an input variable, finding it to be the most important predictor when compared to other traditional indicators. Hu et al. [1] experimented with five machine learning algorithms based on multidimensional value patent portfolios for the high-value patents identification. The Random Forest approach performs the best overall, according to the results. Liu et al. [18] proposed a multi-task learning framework that unifies the identification of high-value patents and standard-essential patents, leveraging the mutual reinforcement of both tasks. The proposed model, which uses both structured and embedded textual features of patents, significantly outperforms single-task learning models with regard to accuracy, recall, precision, and F1 measure across both balanced and imbalanced datasets.

However, a critical challenge remains in these machine learning approaches: the performance heavily depends on hyperparameter tuning, which is often performed manually or through conventional search strategies such as grid search, which are time-consuming and prone to suboptimal solutions. Existing intelligent optimization algorithms used for this purpose, such as Genetic Algorithm (GA) and Particle Swarm Optimization (PSO), still face issues like slow convergence and getting trapped in local optima when applied to real-world patent datasets.

In a patent evaluation, the patent lifetime has been used as a substitute for the commercial potential of a patent. Choi et al. [19] suggest a method to assess the commercial potential of individual patents by applying the feed-forward neural network model to forecast the probability that a patent will remain valid until its maximum expiry date and develop a patent business potential evaluation system at the end. Kumar et al. [20] developed a hybrid model that combines binomial regression and multi-class classification to accurately predict the renewal life of Indian patents, addressing the challenge of unusual renewal life distribution and achieving 90% accuracy.

By comparing multiple solutions until an optimal or satisfying response is discovered that produces a higher accuracy score than the previous one, optimization algorithms are used to increase the efficiency of the machine learning area [21]. Machine learning algorithms often require manual hyperparameter selection to achieve optimal performance. This brings difficulties, including more complexity and the requirement for expert knowledge to tune parameters effectively [22,23]. Intelligent optimization algorithms like Genetic Algorithm and Particle Swarm Optimization can be used to find optimal hyperparameters, improving the efficiency and performance of neural networks without exhaustive manual tuning [24,25,26].

Fire Hawk Optimizer (FHO) is a novel intelligence algorithm proposed in recent years, inspired by the process in nature where Fire Hawks capture prey by setting fires [27]. FHO demonstrates superior performance and exceptional results in addressing structural engineering design problems [28,29]. This paper combined the FHO algorithm and Random Forest and carried out a high-value patents recognition experiment based on the patent data from the Patyee database. The primary goal of this study is to propose an improved optimization algorithm and validate its effectiveness through the representative real-world application of high-value patent recognition. Nevertheless, the canonical FHO algorithm suffers from poor convergence speed and vulnerability to local optima, limiting its effectiveness for complex hyperparameter tuning in machine learning tasks. There is a lack of research focusing on enhancing FHO to overcome these shortcomings in the context of patent data analysis. Regarding the issues of the Fire Hawk optimization algorithm’s poor convergence speed and vulnerability to local optimization [30], an enhanced Fire Hawk optimization algorithm called EFHO is proposed. The EHFO improved the canonical FHO by integrating four strategies of adaptive tent chaotic mapping, hunting prey, adding the inertial weight, and an enhanced flee strategy to address the weakness of FHO development. Extensive benchmark experiments on 23 well-known test functions demonstrate that EFHO achieves faster convergence, higher optimization accuracy, and better stability compared to FHO, PSO, GWO, and WOA. Further tests on large-scale problems (up to 1000 dimensions) confirm EFHO’s robustness and dimension insensitivity. While other intelligent optimization algorithms could potentially be applied for hyperparameter tuning, many face challenges like slow convergence or getting trapped in local optima, which EFHO effectively addresses. Applying EFHO to optimize Random Forest hyperparameters for high-value patent recognition on a real-world patent dataset, the EFHO-RF model outperforms other classifiers in recognition accuracy and classification stability, demonstrating EFHO’s practical effectiveness and broad applicability in this domain. The remainder of this paper is structured in the following way. Section 2 introduces the original Fire Hawk Optimization Algorithm. The enhanced Fire Hawk Optimization Algorithm is presented in Section 3. Simulation experiments are described in Section 4. Section 5 explains the application of identifying high-value patents with Random Forest. Finally, the conclusion is presented in Section 6.

## 2. Fire Hawk Optimization Algorithm

FHO is a novel intelligence optimization algorithm suggested in recent years. The Fire Hawk in the algorithm refers to birds such as brown falcon, black kite and whistling kite. These birds are called Fire Hawk because they find and capture prey by setting fire while hunting in the wild. In the FHO algorithm, the candidate solutions in the search space are mapped to the three categories: the main fire, the prey and the Fire Hawks. The concrete steps of the optimization algorithm are as follows:

(1)Compute the fitness value for all solution candidates. The best solution candidate in the global searching space is presumed to be the main fire, with the better n solution candidates considered as the Fire Hawks, and the remaining solutions referred to as prey.(2)Calculate the length between the prey and the Fire Hawks, and assign each prey to the nearest Fire Hawk based on the distance, thus establishing a territory for each Fire Hawk. The determination of distance relies on the following equation:
(1)$${D_{i}}^{j}=\sqrt{{\left(x_{2}-x_{1}\right)}^{2}+{\left(y_{2}-y_{1}\right)}^{2}},\left\{\begin{array}{c} i=1,2,\ldots ,n\\ j=1,2,\ldots ,m \end{array}\right.,$$
where ${D_{i}}^{j}$ represents the length between the *i*th Fire Hawk and the *j*th prey, m represents the number of prey, n represents the number of Fire Hawks, $\left(x_{1},y_{1}\right)$ represents the coordinate of the Fire Hawk, $\left(x_{2},y_{2}\right)$ represents the coordinate of the prey.
(3)From the main fire, the Fire Hawks gather burning sticks and set fire in their specific territory to force the prey to hastily flee, then move to a new location. Meanwhile, some Fire Hawks are excited to use the sticks that are burning in other Fire Hawks’ territories. The equation of location updating procedures is as follows:
(2)$$FH_{i}^{new}=FH_{i}+(r_{1}\times MF-r_{2}\times FH_{Near}),i=1,2,\ldots ,n$$
where $FH_{i}^{new}$ represents the updated location vector of the *i*th Fire Hawk ($FH_{i}$), MF represents the globally optimal location in the searching space that is regarded as the main fire, $FH_{Near}$ represents another Fire Hawk in the searching space, $r_{1}$ and $r_{2}$ are evenly distributed random integers between 0 and 1 that determine how Fire Hawks will migrate toward the main fire or other occupied territories.
(4)The prey in the Fire Hawk’s territory start to run away once it sets fire; they may run away, hide, or mistakenly run in the direction of the Fire Hawk. During the location update process, these actions can be expressed by the following equation:
(3)$$PR_{q}^{new}=PR_{q}+(r_{3}\times FH_{l}-r_{4}\times SL_{l}),\left\{\begin{array}{c} l=1,2,\ldots ,n.\\ q=1,2,\ldots ,r. \end{array}\right.$$
where $PR_{q}^{new}$ represents the updated location vector of the *q*th prey, which is denoted as ${\mathrm{PR}}_{q}$ that the *l*th Fire Hawk, which is denoted as $FH_{l}$ is encircling, ${\mathrm{SL}}_{l}$ represents a safe location in the territory of $FH_{l}$, $r_{3}$ and $r_{4}$ are evenly distributed random integers between 0 and 1 that are used to determine how prey will move toward the safe location and the Fire Hawks. The mathematical presentation of $SL_{l}$ is formulated as follows:(4)$$SL_{l}=\frac{\sum_{q=1}^{r}PR_{q}}{r},\left\{\begin{array}{c} q=1,2,\ldots ,r\\ l=1,2,\ldots ,n \end{array}\right.$$
where the $PR_{q}$ is the *q*th prey that $FH_{l}$ is encircling.
(5)Additionally, the prey may run out of the current Fire Hawk’s territory, potentially to another Fire Hawk’s territory, or to a safer location. The position update equation takes these actions into consideration.
(5)$$PR_{q}^{new}=PR_{q}+(r_{5}\times FH_{Other}-r_{6}\times SL),\left\{\begin{array}{c} l=1,2,\ldots ,n.\\ q=1,2,\ldots ,r. \end{array}\right.$$
In this equation, $PR_{q}^{new}$ represents the updated location vector of the *q*th prey which is denoted as $PR_{q}$ that the *l*th Fire Hawk which is denoted as $FH_{l}$ is encircling, $FH_{Other}$ represents another Fire Hawk in the searching space, SL represents a safe location beyond the territory of $FH_{l}$, $r_{5}$ and $r_{6}$ are evenly distributed random integers between 0 and 1 that are used to determine how prey will approach the safe location beyond the territory or the other Fire Hawks. The equation of SL is as follows:(6)$$SL=\frac{\sum_{j=1}^{m}PR_{j}}{m}$$
where the $PR_{j}$ is the *j*th prey in the search space.
(6)Return to step 1, loop until certain conditions are met to obtain the global best solution, and the algorithm ends.

A growing body of research has applied FHO to diverse real-world and engineering problems, demonstrating its versatilibiomimetics-10-00561ty and effectiveness. In construction project management, Shishehgarkhaneh et al. [28] utilized FHO within a Building Information Modeling framework to balance multiple resource trade-offs such as time, cost, quality, risk, and environmental impact. Their results highlighted FHO’s capability to yield competitive and exceptional solutions in multi-objective scheduling problems. Similarly, Hosseinzadeh et al. [29] applied FHO to enhance security and energy efficiency in wireless sensor networks through a trust-based routing protocol, achieving superior performance in network reliability and energy consumption.

Algorithmic improvements of FHO itself have also been explored. Ashraf et al. [30] introduced novel swarm initialization techniques leveraging quasi-random sequences to boost convergence rates and diversity, significantly outperforming the standard FHO. Baweja et al. [31] proposed the Levy Flight-based Fire Hawk Optimizer, enhancing exploration and reducing premature convergence, with experimental validations confirming improved optimization outcomes on standard benchmarks.

In the domain of human-computer interaction and activity recognition, Alonazi et al. [32] integrated FHO with deep learning for hyperparameter tuning, resulting in enhanced classification accuracy and robustness for human activity detection. In renewable energy modeling, Said et al. [33] employed a modified version of FHO to precisely extract photovoltaic parameters in solar cell models, outperforming other metaheuristic algorithms and demonstrating superior prediction accuracy. In energy system modeling, Khajuria et al. [34] applied a modified FHO to identify unknown parameters in solid oxide fuel cell models, achieving highly accurate parameter estimation across varying temperatures and pressures.

## 3. Enhanced Fire Hawk Optimization Algorithm

Although recent FHO variants have introduced improvements such as quasi-random sequence initialization [30], Levy flight strategies [31], and domain-specific hybridization [32,33], these methods generally focus on enhancing either the exploration phase or the exploitation phase in isolation. They often fail to simultaneously maintain population diversity and balance global–local search throughout the optimization process, particularly in high-dimensional or complex search spaces.

The EFHO improved the canonical FHO by integrating four strategies of hunting prey, adaptive tent chaos mapping, adding the inertial weight, and an enhanced flee strategy to address the weakness of FHO development. These four strategies are designed to work in a complementary manner, jointly strengthening both exploration and exploitation while preserving population diversity. This holistic enhancement aims to overcome premature convergence and improve robustness across a wider range of problem types.

### 3.1. Adaptive Tent Chaos Mapping Strategy

In the original algorithm FHO, the initialization strategy is to randomly distribute throughout the entire space, which has high randomness and uneven distribution, leading to a lack of population diversity. Introducing an adaptive tent chaotic mapping strategy to provide a more uniform distribution is beneficial for obtaining high-quality initial populations. The following equation represents the adaptive tent chaos mapping, assuming that the sequence’s beginning values are random numbers between 0 and 1 [35].(7)$$s_{i+1}=\left\{\begin{array}{cc} 1-rand(0,1) & s_{i}=0\\ 1-2*s_{i} & 0<s_{i}<0.5\\ 2*(1-s_{i}) & 0.5<s_{i}<1 \end{array}\right.$$In this equation, $s_{i+1}$ represents the *i* + 1th mapping value, and the $s_{i}$ represents the *i*th mapping value. The following is the modified initialization equation.(8)$${X_{j}}^{i}(initial)=X_{j}^{\min}+s_{i}*(X_{j}^{\max}-X_{j}^{\min}),\;\left\{\begin{array}{c} i=1,2,\ldots ,N\\ j=1,2,\ldots ,d \end{array}\right.$$In this equation, $X_{j}^{i}(initial)$ represents the *j*th dimension value of the *i*th object in the initial population, $X_{j}^{\min}$ represents the minimum value on the *j*th dimension, $X_{j}^{\max}$ and represents the maximum value on the *j*th dimension.

### 3.2. Hunting Prey

The FHO algorithm lacked the hunting prey part for Fire Hawk. To broaden the scope of the search and further strengthen the capability of local development, the part of hunting prey was added. After the prey flees, the fitness value is calculated based on the position after fleeing, and the prey that has the best fitness value is selected for hunting. The hunting method refers to the spiral attack strategy of the whale optimization algorithm [36], and the equation is as follows:(9)$$FH_{l}^{new}=D_{l}^{\prime }\times e^{bk}\times \cos(2\pi k)+PR_{l}^{best}$$
where ${\mathrm{PR}}_{l}^{best}$ represents the prey with the best position after fleeing, $D_{l}^{\prime }$ represents the length between the prey (${\mathrm{PR}}_{l}^{best}$) and the Fire Hawk ($FH_{l}$), the parameter b represents the shape parameter of the logarithmic spiral; this paper sets b at 1, and the parameter k represents an arbitrary number between −1 and 1.

### 3.3. Adding the Inertia Weight

This paper adds the inertia weight to the equation for setting fire to the Fire Hawks. The changed equation is as follows:(10)$$FH_{l}^{new}=\omega \times FH_{l}+(r_{1}\times GB-r_{2}\times FH_{Near})$$

The inertia weight plays an important role in enhancing the search accuracy and accelerating the convergence speed of solutions [37,38]. In the early stage, larger inertia weight facilitates a more robust global search capability, whereas in the later stage, smaller inertia weight has stronger exploitation capability [39]. Consequently, it is imperative to adjust the inertia weight in a non-linear decreasing fashion. The following is the equation of inertia weight:(11)$$\omega =\frac{1+\sin(e^{\frac{t}{T}+a})}{b}$$
where T represents the total iteration number, t represents the current iteration number, and a and b are constants (i.e., a = 0.5 and b = 20).

To visually illustrate the features of the inertia weight function suggested in this paper, Figure 1 of this paper shows the function curve of the inertia weight, where the iteration number is shown on the x-axis with a maximum of 50, and it can be observed that the inertia weight gradually decreases from 0.1 to 0 as the iteration number increases.

### 3.4. Enhanced Flee Strategy

There are two flee strategies in the original algorithm, where all prey implemented both flee strategies in sequence. In order to increase population diversity, some prey implemented the first flee strategy while others implemented the second flee strategy. The Equations (3) and (5) are combined as follows:(12)$$PR_{q}^{new}=\left\{\begin{array}{cc} PR_{q}+(r_{3}\times FH_{l}-r_{4}\times SP_{l}) & \tau \leq 0.5\\ PR_{q}+(r_{5}\times FH_{Alter}-r_{6}\times SP) & \tau >0.5 \end{array}\right.$$
where τ is evenly distributed random integers between 0 and 1.

In addition, this paper introduces the Levy flight [40,41] for the first flee strategy and the t-distribution perturbation [42,43] strategy for the second flee strategy. The equation of the Levy flight for the first flee strategy is as follows:(13)$$PR_{flee1}^{L}=PR_{flee1}^{best}\times levy(D)$$
where $PR_{flee1}^{best}$ is the best solution candidate in the prey’s implemented first flee strategy, levy(D) represents the Levy flight function, and D is the dimension. After the levy flight, a greedy selection is made between the old and new positions, choosing the one that has better fitness value for the next iteration. The following is the greedy selection equation:(14)$$PR_{flee1}^{best}=\left\{\begin{array}{c} PR_{flee1}^{best}\;\;\;f(PR_{flee1}^{best})\;<\;f(PR_{flee1}^{L})\\ PR_{flee1}^{L}\;\;f(PR_{flee1}^{best})\;\geq \;f(PR_{flee1}^{L}) \end{array}\right.$$

Levy flying, which combines random movement over long and short distances, is a technique to imitate the random flight of animals. During the initial phase of the algorithm, long distances can expand the search range and explore discoveries, which is beneficial for enhancing population diversity and reducing the risk of local optima. In the latter stage of the algorithm, the range of the global optimal solution is basically determined, and short distances can improve the accuracy of the solutions, enabling the algorithm to converge to the global best.

The equation of t-distribution perturbation for the second flee strategy is as follows:(15)$$PR_{flee2}^{T}=PR_{flee2}^{best}\times (1+t\_distribution(t))$$
where $PR_{flee2}^{best}$ is the best solution candidate in the prey that implemented the second flee strategy. The t-distribution function is a distribution function that is symmetric about the y-axis, and its two bounding distributions are the standard Cauchy and Gaussian distributions; this function usually contains only one parameter, called degrees of freedom. The degrees of freedom are the iteration number t in this paper, and the following is the equation of the probability density function.(16)$$f(x)=\frac{\Gamma (\frac{t+1}{2})}{\sqrt{\pi t}\Gamma \left(\frac{t}{2}\right)}{\left(1+\frac{x^{2}}{t}\right)}^{-\frac{t+1}{2}},\;-\infty <x<+\infty$$

In the early stage with a small iteration number, the t-distribution gains the ability by approaching the Cauchy distribution. In the middle of the iteration, the t-distribution balances the convergence and population variety of the algorithm by falling between the Gaussian distribution and the Cauchy distribution. In the later stage, the t-distribution gets close to the Gaussian distribution as the iteration number is large, which facilitates the local exploration ability. After the perturbation, the greedy selection model is also used:(17)$$PR_{flee2}^{best}=\left\{\begin{array}{c} {\mathrm{PR}}_{flee2}^{best}\;\;\;f(PR_{flee2}^{best})\;<\;f(PR_{flee2}^{L})\\ PR_{flee2}^{L}\;\;f(PR_{flee2}^{best})\;\geq \;f(PR_{flee2}^{L}) \end{array}\right.$$

When the prey flees, the algorithm performs local exploitation around the fire hawk. As the iteration number increases, most of the prey may stay around the fire hawk, causing the algorithm to stagnate and be unable to further solve the global optimal solution. During the flee stage of prey, using the t-distribution perturbation strategy on the current location can prevent prey from staying around the fire hawk and reduce the likelihood of algorithmic stagnation. The convergence speed and the capacity to deviate from local optima were further improved.

The following is a description of the EFHO algorithm steps:

Step 1: Initialization.

The dimension Dim, maximum iterations of number T, and population size pop are set. The population is initialized by Equation (8). Set the current iteration number t as 1.

Step 2: Evaluation.

Compute fitness value for all individuals, the individual that has the best fitness value is considered as the main fire, select better n individuals as the Fire Hawks, make the remaining individuals as prey.

Step 3: Determining Territory.

For each Fire Hawk, compute the length between each Fire Hawk and each prey by Equation (1), and determine its territory where the preys belong.

Step 4: Setting Fire.

The Fire Hawks set fire and adjust the position according to Equation (10).

Step 5: Prey flees.

Some prey flee by the first strategy by Equation (12) and calculate the fitness for each prey. The prey with the best fitness value after fleeing then implements the Levy flight according to Equations (13) and (14). Some prey flee by the second strategy and calculate the fitness for each prey. The prey with the best fitness value after fleeing then implements the t-distribution perturbation according to Equations (15) and (17).

Step 6: Hunting prey.

The Fire Hawks hunt prey and adjust their position according to Equation (9).

Step 7: Termination.

When the termination condition is reached, the optimal individual position and fitness value are outputted, and the algorithm terminates; if not, proceed to Step 2.

## 4. Simulation Experiment

Two experiments are conducted in this section. The first experiment tested the capability of the EFHO algorithm by comparing it with several closely related algorithms, and the second experiment tested the scalability of the enhanced algorithm for large-scale problems.

### 4.1. Experiment Setup

To evaluate the proposed algorithm’s convergence rate and optimization precision, 23 well-known test functions of CEC2005 [44] are used for comparison experiments. In these functions, F1 to F7 are unimodal and commonly used as benchmark functions for evaluating search accuracy. Optimization algorithms often struggle to converge to the global optimum for these functions. F8 to F13 are multi-modal functions used to assess global search performance and have multiple local extreme points. F14 to F23 are benchmark functions that are fixed-dimension multi-modal. To obtain unbiased results, the maximum iteration number was set to 500 and the population size was set to 30, for all functions in all comparisons. For each function, all algorithms were independently run 30 times.

All experiments were carried out on a personal computer with Intel(R) Core(TM) i9-12900H CPU, 32.00 GB RAM and Windows 11 operating system, and MATLAB R2023b was used to implement all algorithms.

### 4.2. Performance Comparison

In this paper, FHO, PSO (Particle Swarm Optimization), GWO (Grey Wolf Optimization) and WOA (Whale Optimization Algorithm) are selected for comparison with EFHO to evaluate the performance of EFHO. Since some of the 23 functions do not have an optimal value of 0, this paper calculates fitness by taking the absolute value of the discrepancy between the computed result and the actual optimal value of the function. This approach converts all optimal values to 0. This paper uses the best value (Best), mean value (Mean), and standard deviation (Std) to evaluate the performance of all algorithms. Table 1 displays the test results; the best result for each function is boldfaced.

According to Table 1, for F1, F2, F3, F4, F9 and F11, EFHO can achieve the ideal value in theory. For F5, F6, F7, F8 and F13, the outcomes of EFHO outperform the other four algorithms. For F10, the best values obtained by the EFHO algorithm are the same as those of the FHO algorithm, superior to the other three algorithms. The above results demonstrate the remarkable accuracy and stability of EFHO.

To assess the statistical significance of the performance differences between EFHO and the baseline algorithms, the Wilcoxon signed-rank test was conducted for each of the 23 benchmark functions. Table 2 summarizes the number and proportion of functions where EFHO achieved a significantly better result (*p* < 0.05) compared to each baseline. The detailed test results are provided in Appendix A.

As shown in Table 2, EFHO achieved statistically significant improvements (*p* < 0.05) over PSO, GWO, and WOA in more than half of the benchmark functions, and outperformed FHO in 39.13% of the cases. While the proportion of significant wins varies across comparisons, these results still indicate that EFHO consistently delivers competitive or superior performance against the baseline algorithms across a wide range of test functions.

Figure 2 displays the five algorithms’ convergence curves for each test function, which better illustrates the benefit of EFHO. It is demonstrated that among them, the EFHO convergence rate is relatively quick. The EFHO convergence curves are smooth and decline quickly for the majority of functions.

In conclusion, the EFHO algorithm demonstrated superior convergence, stability, and accuracy compared to the original FHO and the other four algorithms, which adequately validates the feasibility of the suggested method for FHO improvement in this paper.

### 4.3. EFHO’s Scalability Test for Large-Scale Problems

This paper further confirms whether the EFHO is scalable when it comes to solving large-scale problems, as real-world engineering applications often encounter large-scale optimization challenges. Because the dimensions of F14 to F23 are fixed, the paper selects test functions F1 to F13 and respectively sets the dimensionality of EFHO to 500 and 1000. Table 3 displays the high-dimensional experiment results.

Table 3 shows that even for functions with 500 and 1000 dimensions, EFHO still has a good accuracy. Especially for F1, F2, F3, F4, F9 and F11, EFHO can achieve the ideal value 0 in theory. The results of EFHO for other functions are in line with those of 500 and 1000 dimensions, suggesting that EHO is not dimension-sensitive during problem-solving. The above analysis shows that EFHO performs well when handling large-scale problems since it is not significantly impacted by a significant increase in dimension while maintaining high accuracy.

## 5. High-Value Patents Recognition with Random Forest

Leo Breiman and Adele Cutler first presented the Random Forest ensemble learning method in 2001 [45]. It addresses classification and regression problems by constructing multiple decision trees and then improving prediction accuracy, generalization ability, and resistance to overfitting through averaging (for regression problems) or majority voting (for classification problems) [46]. Random Forest’s basic idea is to construct many decision trees, where each tree is independent and the features within each tree are randomly selected, thereby reducing the model’s variance [47]. To get the final prediction result, Random Forest averages or votes on each tree’s prediction results. Random Forest works well for a variety of regression and classification problems, especially for complex, high-dimensional datasets and problems that require handling a large number of features. Due to its excellent generalization ability and resistance to overfitting, Random Forest performs well in many practical applications [48].

The Random Forest model has a number of hyperparameters that can be tuned by the user to optimise its performance, such as the number of trees, the minimum number of samples required in a node, and the number of features to be considered for each segmentation. These hyperparameters, also called tuning parameters, control the construction and training process of the model. Because optimal tuning parameter values depend on the dataset being used, they must be carefully chosen [49]. Many researchers employ intelligent optimization algorithms to tune the hyperparameters of random forests, seeking the best hyperparameters for the current dataset [50,51,52]. This paper employs the EFHO algorithm to optimize the hyperparameters of the random forest and conducts application research for the high-value patents identification based on it.

### 5.1. Dataset

The patent dataset used in this paper is collected from the Patyee database. Patyee is a comprehensive database providing extensive information on patents, including over 180 million in-depth processed patent records from more than 171 countries, regions, and organizations worldwide. This paper retrieves Chinese invention patents that have received awards, such as the China Patent Award, have been published for over ten years, and are still valid, using them as positive samples for the high-value patents dataset, totaling 10,638 patents. The choice of awarded patents as high-value patents is based on the award attribute provided by the Patyee database, which compiles official records from the China National Intellectual Property Administration and other authoritative sources, and is supported by prior research [5,18] with two studies in our references also adopting this approach. Given that such awards are granted through rigorous expert review, the accuracy of this labeling is expected to be very high.

Additionally, to construct the negative sample set, this study retrieves Chinese invention patents with a patent value of 100, published for over ten years, and still valid, totaling 18,030 patents. After removing the awarded patents, the remaining 17,734 patents are used as negative samples. The choice of value 100 is mainly driven by two considerations. First, the number of non-awarded patents is extremely large, reaching several million, making it necessary to narrow the search with specific criteria. Selecting those with a value of 100 results in a set size close to that of the positive samples, which facilitates a relatively balanced dataset. Second, the value score is an internal evaluation index calculated by the Patyee database using a proprietary method that is not publicly disclosed. A score of 100 is simply one of the discrete values in this system and is used here as a filtering attribute, not as a classification threshold. While such database-calculated values do not necessarily indicate truly high-value patents, awarded patents are determined through expert review and can be considered genuinely high-value. In this study, patents with a value of 100 but without awards are treated as negative samples, not to imply that they have low value, but to establish a clear and consistent labeling criterion in which only awarded patents are considered positive. This approach minimizes ambiguity in the definition of high-value patents and avoids dependence on database-specific scoring methods, ensuring consistency in the training labels. Additionally, this paper selects both positive and negative samples from patents that have been published for over ten years, in order to exclude the influence of the publication time.

Numerous studies on patent value indexes have been conducted, taking into account various scales and embracing a wide range of perspectives [53]. In paper 18, researchers used the title and abstract of patents as textual features, which were converted into vectors using the BERT model and then concatenated with other structural features to serve as the model’s input for training. The final accuracy was only a little over sixty percent, which is lower than the accuracy obtained by using only structural features for training [1]. This indicates that textual features contribute little to the identification of high-value patents and might even act as noise. Therefore, this paper does not use textual features but selects the most commonly used patent value indicators as the features of the dataset. Table 4 lists the 14 structural features that are present in the dataset.

### 5.2. Data Preprocessing

An essential stage in machine learning is data preprocessing. Enhancing data quality, eliminating noise and outliers, and guaranteeing consistency and completeness involves cleaning, converting, and standardizing data.

Missing data can lead to biased results, reduce the statistical power of analyses, and affect the overall performance of machine learning models. There are two approaches to handling missing data: deletion or imputation. There are many methods for imputation, with the most common being replacing missing values with the mean or median. This paper firstly performed a missing data check on the dataset and found that every row and column contains data, so no missing value treatment is necessary.

The dataset used in this paper was split 80:20, with the training set being the remaining 80% and the testing set being 20%. Prior to the split, the dataset was randomly shuffled to ensure a balanced distribution of the data. Before training, this paper also performed normalization on the data, transforming the values of each feature to a value between 0 and 1. The equation of normalization is as follows:(18)$$v_{norm}=\frac{v-v_{\min}}{v_{\max}-v_{\min}}$$
where v represents the raw value. $v_{\min}$ represents the minimum value, $v_{\max}$ represents the maximum value. $v_{norm}$ represents the transformed value after normalization, which will be within the range of 0 to 1.

### 5.3. Experiments and Results

Experiment Setup

This paper optimized the Random Forest using the EFHO algorithm and conducted experimental verification based on the patent dataset. This paper also selects Naive Bayes (NB), Back Propagation Neural Network (BP), Support Vector Machine (SVM), and Logistic Regression (LR) for comparison. This article uses MATLAB for experiments; each model was implemented using built-in MATLAB functions. The NB model was called using the fitcnb function, the LR model using the fitclinear function, the SVM model using the fitcsvm function, the RF model using the TreeBagger function, and the BP neural network using the feedforwardnet function; the hidden layer’s node number is set to 8.

In this experiment, the objective function is the mean square error of the classification result, and the formula for the mean square error is as follows:(19)$$M=\frac{1}{n}\sum_{i=1}^{n}{\left(x_{i}-{\hat{x}}_{i}\right)}^{2}$$
where n represents the amount of data in the training or testing set, $x_{i}$ represents the true value, and ${\hat{x}}_{i}$ represents the predicted value of the model. The algorithm terminates when either the maximum iteration number is exceeded or the objective function falls below the specified threshold.

2.Results

This paper separately applies the LR model, the NB model, the RF model, the SVM model, the BP model and the RF model optimized using EFHO (EFHO-RF) to high-value patents recognition. To compare the optimization effects, this paper also optimized the BP model using the EFHO algorithm as EFHO-BP. Table 5 displays the results of the experiment. Five-fold cross-validation was used to evaluate each model by indicators such as accuracy, recall, precision, F1 measure and AUC. The AUC means the Area Under the Curve (AUC) of the Receiver Operating Characteristics (ROC) curve.

According to the results of the experiment, the EFHO-RF model performs best, with all indicators, except for precision, being at the highest level. All indicators are superior to the pre-optimization RF model. The accuracy improved from 95.8% before optimization to 96.6%, the recall improved from 91.8% to 96.3%, the precision increased from 96.7% to 97%, the F1 measure increased from 94.2% to 96.3%, and the AUC improved from 98.8% to 99%. The EFHO-BP also performed well, with accuracy, recall, and F1 measure surpassing all models except EFHO-RF. However, its precision and AUC were lower than those of the RF model before optimization, and its precision was even lower than that of the BP model before optimization. The experimental results indicate that EFHO can find better hyperparameters for the Random Forest, providing a desirable solution for the problem. In this classification task, EFHO contributed to improving the Random Forest’s use of 14 patent-level indicators (Table 4) by identifying hyperparameter settings that better balance feature relevance and model complexity. While a separate feature importance ranking was not conducted, the selected features are widely recognized in patent analytics as being strongly associated with patent value, based on both the literature and our long-term practical experience.

In order to assess the discriminative ability of the model, this paper also examined the ROC curve. The AUC for the optimized Random Forest was 99.0%, demonstrating a high level of performance. The ROC curve shows a marked increase in true positive rates while keeping low false positive rates, further validating the robustness of our optimization approach. These results highlight the potential of combining metaheuristic algorithms like FHO with machine learning methods to enhance classification tasks, providing a more reliable and efficient model for practical applications. The ROC curves before and after optimization are shown in Figure 3. This suggests that the Random Forest model’s classification performance is excellent, and the EFHO algorithm can help improve the classification performance.

To make the identification results more transparent, this paper further conducted a feature importance analysis using the out-of-bag (OOB) permutation method. The results are shown in Figure 4. As displayed, the claims number (CLMSN), the countries number belongs to the same family (NCSE), and the assignments number (RAQ) ranked as the top three most influential features. This highlights the central role of patent scope, international family coverage, and technology transfer activities in distinguishing high-value patents. By contrast, features such as the reexamination number (NR) and the applicant’s number (PAN) showed limited or negligible importance in classification. These findings indicate that the EFHO-RF model identifies high-value patents by leveraging a meaningful combination of legal, citation, and family-related attributes. Presenting feature importance thus makes the identified results more visible and interpretable, complementing the performance evaluation and providing additional insights into the determinants of patent value.

## 6. Conclusions

This paper introduces a novel enhanced Fire Hawk optimization algorithm called EFHO. The EFHO improves upon the canonical FHO by incorporating four key strategies: adaptive tent chaotic mapping, hunting prey, the addition of inertial weight, and an enhanced flee strategy. These modifications address the shortcomings of the original FHO, such as weak convergence and limited exploration capabilities. By enhancing these aspects, EFHO provides a more robust and efficient optimization method.

While EFHO is a general intelligent optimization algorithm, this study uses high-value patent recognition primarily as a representative application scenario to validate its effectiveness. The challenges inherent in such real-world applications demand optimization algorithms with faster convergence and stronger global search ability than traditional methods. Existing hyperparameter tuning techniques often suffer from inefficiency and suboptimal solutions in such scenarios.

This paper used the EFHO algorithm to optimize the Random Forest model’s hyperparameters in the chosen application scenario. The resulting EFHO-RF classification model leverages the strengths of both EFHO and Random Forest, leading to superior performance. EFHO effectively tuned key Random Forest hyperparameters such as the number of trees, the minimum number of samples required in a node, and the number of features to be considered for each segmentation, leading to better utilization of informative features and reduced overfitting. The optimization process improved the model’s balance between feature relevance and complexity, resulting in more stable and accurate classification outcomes. Experimental results demonstrate that this model achieves a high level of accuracy on the selected patent dataset, significantly outperforming traditional methods. As awarded patents are determined through expert review, the identified positives inherently represent truly valuable patents, aligning with the ultimate goal of high-value patent identification. Thus, this research not only advances optimization algorithm methodology but also demonstrates its applicability through a representative real-world task, highlighting the method’s relevance and effectiveness in applied settings.

Beyond the empirical results, this study offers broader scientific and practical value. The proposed EFHO-RF framework provides a replicable approach for addressing other machine learning tasks. Its successful validation in the chosen application scenario offers methodological insights for similar applied analytics and decision-support contexts.

However, certain limitations should be acknowledged. The dataset is limited to Chinese invention patents, and the positive samples are defined solely by award status, which, although highly accurate, may not fully capture all possible interpretations of patent value. In addition, the EFHO-RF model has yet to be tested on datasets from other jurisdictions or with different feature spaces, which may affect its generalizability. It should also be noted that high-value patent recognition is essentially a complex classification problem. By leveraging EFHO’s strong capability in hyperparameter optimization and Random Forest’s robust classification ability, the EFHO-RF framework provides a suitable and effective solution for this type of task. Future work will include cross-domain validation, incorporation of alternative indicators of patent value, and application of EFHO to diverse and heterogeneous datasets.

## Figures and Tables

**Figure 1 biomimetics-10-00561-f001:**
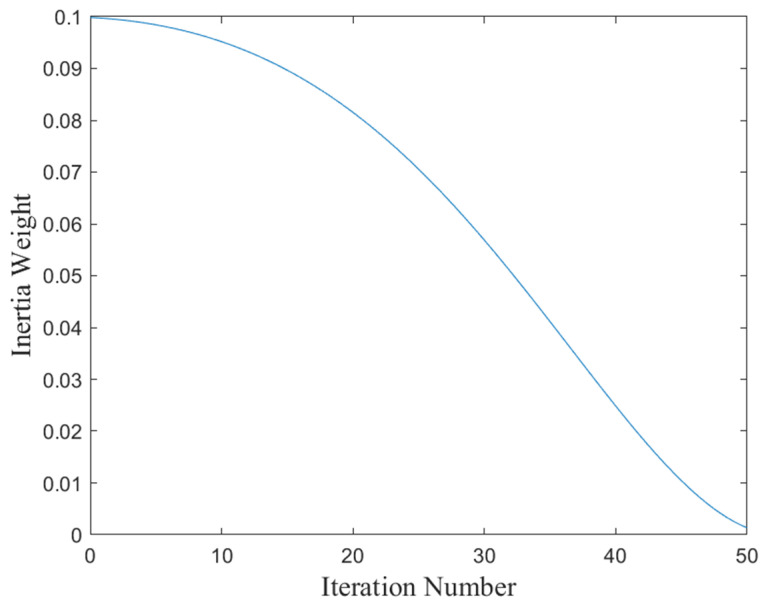
The function curve for inertia weight.

**Figure 2 biomimetics-10-00561-f002:**
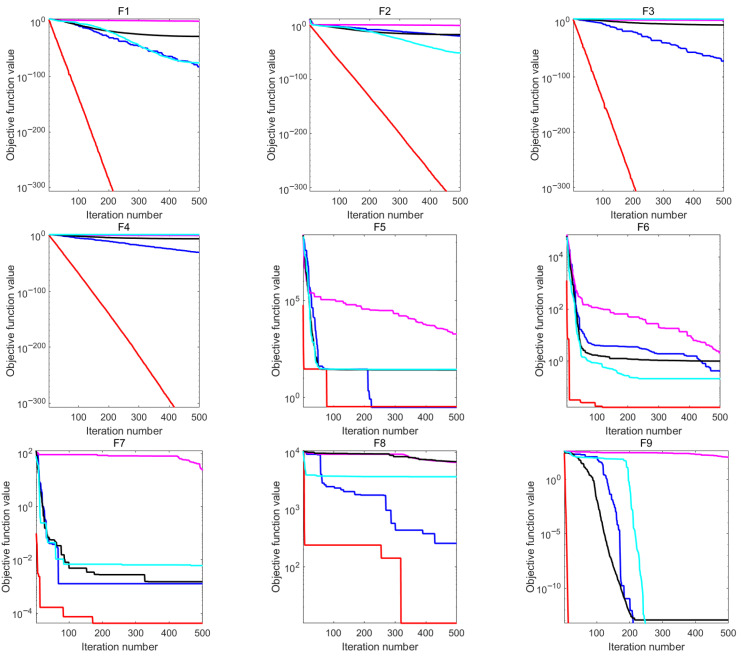
The convergence diagram of five algorithms.

**Figure 3 biomimetics-10-00561-f003:**
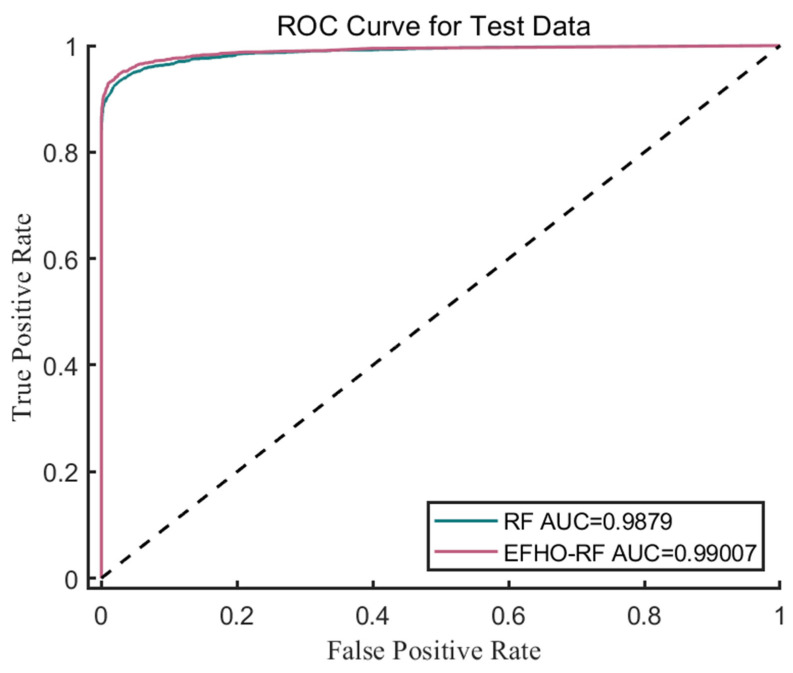
The ROC Curve before and after optimization by EFHO and the dashed line represents the random classifier baseline (AUC = 0.5).

**Figure 4 biomimetics-10-00561-f004:**
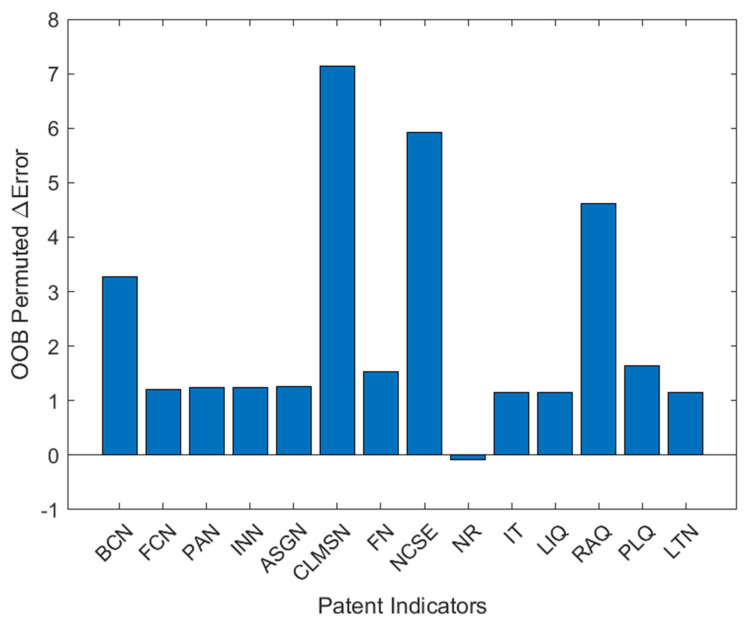
Feature importance of patent indicators in the EFHO-RF model.

**Table 1 biomimetics-10-00561-t001:** The performance comparison results.

	Results	PSO	FHO	GWO	WOA	EFHO
F1	Best	5.86 × 10^−1^	1.4875 × 10^−83^	6.5908 × 10^−29^	1.0343 × 10^−84^	0
	Mean	2.17 × 10^0^	1.3582 × 10^−69^	1.1309 × 10^−27^	7.9071 × 10^−73^	0
	Std	8.76 × 10^−1^	7.0382 × 10^−69^	1.5143 × 10^−27^	4.1528 × 10^−72^	0
F2	Best	2.69 × 10^0^	3.0352 × 10^−21^	1.5109 × 10^−17^	5.31 × 10^−58^	0
	Mean	4.52 × 10^0^	2.4074 × 10^−19^	8.9556 × 10^−17^	1.0498 × 10^−50^	0
	Std	1.24 × 10^0^	3.3718 × 10^−19^	5.7898 × 10^−17^	4.4941 × 10^−50^	0
F3	Best	6.92 × 10^1^	5.4193 × 10^−82^	9.7792 × 10^−9^	1.59 × 10^4^	0
	Mean	1.83 × 10^2^	1.6619 × 10^−69^	4.6808 × 10^−6^	4.38 × 10^4^	0
	Std	5.94 × 10^1^	5.1581 × 10^−69^	7.6958 × 10^−5^	1.40 × 10^4^	0
F4	Best	1.48 × 10^0^	3.2607 × 10^−35^	1.2102 × 10^−7^	1.78 × 10^0^	0
	Mean	2.06 × 10^0^	1.0124 × 10^−30^	7.4167 × 10^−7^	5.33 × 10^1^	0
	Std	2.71 × 10^−1^	3.5189 × 10^−30^	6.7046 × 10^−7^	2.79 × 10^1^	0
F5	Best	2.79 × 10^2^	3.52 × 10^−2^	2.61 × 10^1^	2.71 × 10^1^	2.11 × 10^−3^
	Mean	1.10 × 10^3^	2.80 × 10^−1^	2.71 × 10^1^	2.80 × 10^1^	9.58 × 10^−2^
	Std	7.26 × 10^2^	2.09 × 10^−1^	7.51 × 10^−1^	4.92 × 10^−1^	9.21 × 10^−2^
F6	Best	9.05 × 10^−1^	7.85 × 10^−3^	2.50 × 10^−1^	6.70 × 10^−2^	5.21 × 10^−3^
	Mean	2.26 × 10^0^	8.94 × 10^−1^	8.35 × 10^−1^	3.75 × 10^−1^	1.48 × 10^−2^
	Std	8.27 × 10^−1^	1.66 × 10^0^	3.20 × 10^−1^	2.09 × 10^−1^	8.00 × 10^−3^
F7	Best	2.83 × 10^0^	2.30 × 10^−4^	2.43 × 10^−4^	6.7084 × 10^−5^	5.5551 × 10^−7^
	Mean	1.67 × 10^1^	9.06 × 10^−4^	1.70 × 10^−3^	2.14 × 10^−3^	3.6111 × 10^−5^
	Std	1.37 × 10^1^	4.29 × 10^−4^	9.03 × 10^−4^	1.99 × 10^−3^	3.6888 × 10^−5^
F8	Best	4.31 × 10^3^	3.61 × 10^0^	4.89 × 10^3^	6.76 × 10^−1^	1.34 × 10^−2^
	Mean	6.40 × 10^3^	3.93 × 10^2^	6.54 × 10^3^	2.32 × 10^3^	2.61 × 10^1^
	Std	1.46 × 10^3^	7.02 × 10^2^	6.88 × 10^2^	1.73 × 10^3^	3.40 × 10^1^
F9	Best	9.46 × 10^1^	0	5.6843 × 10^−14^	0	0
	Mean	1.60 × 10^2^	0	1.93 × 10^0^	0	0
	Std	3.98 × 10^1^	0	3.31 × 10^0^	0	0
F10	Best	2.06 × 10^0^	4.4409 × 10^−16^	7.1498 × 10^−14^	4.4409 × 10^−16^	4.4409 × 10^−16^
	Mean	2.78 × 10^0^	4.4409 × 10^−16^	9.6959 × 10^−14^	3.4047 × 10^−15^	4.4409 × 10^−16^
	Std	3.75 × 10^−1^	0 × 10^0^	1.4564 × 10^−14^	2.2625 × 10^−15^	0
F11	Best	3.07 × 10^−2^	0	0	0	0
	Mean	1.48 × 10^−1^	0	5.12 × 10^−3^	0	0
	Std	4.71 × 10^−2^	0	8.16 × 10^−3^	0	0
F12	Best	5.78 × 10^−3^	4.95 × 10^−4^	6.43 × 10^−3^	5.25 × 10^−3^	3.53 × 10^−3^
	Mean	3.31 × 10^−2^	1.47 × 10^−3^	4.08 × 10^−2^	2.31 × 10^−2^	2.35 × 10^−2^
	Std	2.57 × 10^−2^	6.48 × 10^−4^	2.23 × 10^−2^	1.13 × 10^−2^	6.51 × 10^−3^
F13	Best	2.26 × 10^−1^	3.96 × 10^−3^	4.08 × 10^−1^	1.19 × 10^−1^	7.9238 × 10^−9^
	Mean	5.11 × 10^−1^	1.08 × 10^−2^	6.87 × 10^−1^	5.22 × 10^−1^	5.9291 × 10^−7^
	Std	2.13 × 10^−1^	4.94 × 10^−3^	2.38 × 10^−1^	2.44 × 10^−1^	6.9933 × 10^−7^
F14	Best	0 × 10^0^	1.7555 × 10^−8^	1.8985 × 10^−11^	1.0154 × 10^−11^	1.98 × 10^0^
	Mean	2.00 × 10^0^	5.71 × 10^−1^	3.46 × 10^0^	1.83 × 10^0^	6.12 × 10^0^
	Std	2.70 × 10^0^	6.20 × 10^−1^	4.35 × 10^0^	2.96 × 10^0^	1.31 × 10^0^
F15	Best	1.34 × 10^−4^	1.3838 × 10^−5^	6.009 × 10^−10^	1.667 × 10^−6^	1.06 × 10^−4^
	Mean	5.91 × 10^−4^	9.51 × 10^−4^	2.71 × 10^−3^	3.87 × 10^−4^	9.25 × 10^−4^
	Std	1.48 × 10^−4^	1.94 × 10^−3^	6.80 × 10^−3^	3.76 × 10^−4^	6.42 × 10^−4^
F16	Best	2.2204 × 10^−15^	2.1008 × 10^−7^	3.8243 × 10^−10^	6.9722 × 10^−14^	7.1983 × 10^−6^
	Mean	2.4351 × 10^−15^	9.9372 × 10^−6^	2.3649 × 10^−8^	1.1901 × 10^−9^	3.53 × 10^−3^
	Std	3.9858 × 10^−17^	1.0288 × 10^−5^	2.6884 × 10^−8^	4.2983 × 10^−9^	3.65 × 10^−3^
F17	Best	1.6653 × 10^−16^	2.4405 × 10^−5^	7.4074 × 10^−8^	1.7071 × 10^−11^	4.7274 × 10^−5^
	Mean	1.6653 × 10^−16^	3.77 × 10^−4^	2.1562 × 10^−6^	5.8586 × 10^−6^	6.94 × 10^−3^
	Std	0 × 10^0^	3.50 × 10^−4^	2.0217 × 10^−6^	1.1906 × 10^−5^	7.36 × 10^−3^
F18	Best	2.2204 × 10^−15^	1.2711 × 10^−5^	1.4177 × 10^−7^	7.6656 × 10^−8^	3.0105 × 10^−6^
	Mean	1.0214 × 10^−14^	1.23 × 10^−3^	3.0221 × 10^−5^	5.6182 × 10^−5^	1.03 × 10^−1^
	Std	4.704 × 10^−15^	1.08 × 10^−3^	4.3207 × 10^−5^	1.01 × 10^−4^	3.54 × 10^−1^
F19	Best	3.9968 × 10^−15^	2.34 × 10^−4^	1.9841 × 10^−6^	3.8805 × 10^−7^	6.17 × 10^−3^
	Mean	5.1514 × 10^−15^	1.59 × 10^−2^	1.09 × 10^−3^	5.90 × 10^−3^	1.23 × 10^−1^
	Std	1.907 × 10^−15^	5.06 × 10^−2^	2.22 × 10^−3^	7.07 × 10^−3^	1.32 × 10^−1^
F20	Best	7.0895 × 10^−10^	1.96 × 10^−2^	1.582 × 10^−6^	5.8291 × 10^−5^	3.13 × 10^−1^
	Mean	3.96 × 10^−2^	1.25 × 10^−1^	7.13 × 10^−2^	8.22 × 10^−2^	7.69 × 10^−1^
	Std	5.60 × 10^−2^	1.11 × 10^−1^	8.25 × 10^−2^	9.90 × 10^−2^	3.33 × 10^−1^
F21	Best	1.7764 × 10^−15^	1.07 × 10^−1^	4.06 × 10^−4^	4.41 × 10^−4^	1.77 × 10^−4^
	Mean	3.34 × 10^0^	3.78 × 10^−1^	1.10 × 10^0^	2.30 × 10^0^	8.31 × 10^−1^
	Std	3.44 × 10^0^	1.93 × 10^−1^	2.22 × 10^0^	2.86 × 10^0^	7.97 × 10^−1^
F22	Best	0 × 10^0^	7.39 × 10^−2^	4.29 × 10^−4^	3.47 × 10^−4^	1.98 × 10^−2^
	Mean	7.87 × 10^−1^	4.85 × 10^−1^	1.73 × 10^−3^	2.09 × 10^0^	8.65 × 10^−1^
	Std	2.03 × 10^0^	2.23 × 10^−1^	7.32 × 10^−4^	2.74 × 10^0^	7.89 × 10^−1^
F23	Best	8.3489 × 10^−14^	1.13 × 10^−1^	9.8691 × 10^−5^	1.20 × 10^−3^	1.08 × 10^−2^
	Mean	1.24 × 10^0^	5.42 × 10^−1^	1.82 × 10^−1^	2.81 × 10^0^	1.12 × 10^0^
	Std	2.51 × 10^0^	2.95 × 10^−1^	9.70 × 10^−1^	3.25 × 10^0^	9.19 × 10^−1^

**Table 2 biomimetics-10-00561-t002:** Summary of Wilcoxon signed-rank test results.

Comparison	Functions with *p* < 0.05 (n/23)	Proportion (%)
EFHO vs. FHO	9/23	39.13
EFHO vs. PSO	14/23	60.87
EFHO vs. GWO	13/23	56.52
EFHO vs. WOA	12/23	52.17

**Table 3 biomimetics-10-00561-t003:** The EFHO test results of high-dimensional problems.

	EFHO			
	500 Dimensions		1000 Dimensions	
	Mean	Std	Mean	Std
F1	0	0	0	0
F2	0	0	0	0
F3	0	0	0	0
F4	0	0	0	0
F5	7.76 × 10^0^	4.09 × 10^1^	9.48 × 10^−2^	1.28 × 10^−1^
F6	6.61 × 10^−3^	6.66 × 10^−3^	1.72 × 10^−2^	9.96 × 10^−3^
F7	3.3296 × 10^−5^	4.4944 × 10^−5^	2.9348 × 10^−5^	2.603 × 10^−5^
F8	1.65 × 10^1^	1.18 × 10^1^	2.55 × 10^1^	3.47 × 10^1^
F9	0	0	0	0
F10	4.4409 × 10^−16^	0	4.4409 × 10^−16^	0
F11	0	0	0	0
F12	1.62 × 10^−2^	4.78 × 10^−3^	2.64 × 10^−2^	5.58 × 10^−3^
F13	5.5754 × 10^−6^	1.2471 × 10^−5^	2.8439 × 10^−7^	4.072 × 10^−7^

**Table 4 biomimetics-10-00561-t004:** The selected patent value indicators.

No.	Abbr.	Description
1	BCN	The backward citation number
2	FCN	The forward citation number
3	PAN	The applicants number
4	INN	The inventors number
5	ASGN	The patent holders number
6	CLMSN	The claims number
7	FN	The patent family number
8	NCSE	The country’s number belongs to the same family
9	NR	The reexaminations number
10	IT	The invalidation times
11	LIQ	The licensing frequency
12	RAQ	The assignments number
13	PLQ	The pledging frequency
14	LTN	The lawsuits number

**Table 5 biomimetics-10-00561-t005:** Test performance comparison.

Algorithm	Accuracy	Recall	Precision	F1 Measure	AUC
NB	0.918	0.910	0.874	0.891	0.950
LR	0.950	0.898	0.953	0.925	0.966
RF	0.958	0.918	0.967	0.942	0.988
SVM	0.943	0.880	0.964	0.920	0.977
BP	0.947	0.883	0.971	0.925	0.966
EFHO-BP	0.959	0.950	0.963	0.960	0.985
EFHO-RF	0.966	0.958	0.970	0.963	0.990

## Data Availability

The datasets generated or analyzed during this study are available from the corresponding author on reasonable request.

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
