# Peer review of "High-Value Patents Recognition with Random Forest and Enhanced Fire Hawk Optimization Algorithm"

_biomimetics, 2025, doi:10.3390/biomimetics10090561_

Round 1
Reviewer 1 Report
Comments and Suggestions for Authors
- In the abstract and introduction, the authors only emphasize the importance of high-value patent recognition but lack a description of the research gaps related to methods for identifying high-value patents.
- In the abstract and introduction, the authors should expand their discussion to clearly explain why EFHO was chosen for high-value patent recognition, including its necessity and effectiveness. Additionally, the authors should address whether other intelligent optimization algorithms could achieve similar results—i.e., the applicability of the proposed method in high-value patent recognition research.
- In the abstract and introduction, the description of the research results is overly simplistic. If the proposed new method improves recognition accuracy and effectiveness, the authors should further specify the extent of improvement and its core advantages over other methods to highlight the value of this study.
- The authors should provide a systematic review of the applications of the FHO method rather than simply introducing the FHO method itself.
- Provide the full names of PSO, GWO, and WOA.
- Table 1 spans multiple pages; it is recommended to adjust the format to display it on a single page.
- The authors propose the EFHO algorithm, but the process does not address the problem of high-value patent recognition, focusing more on methodological innovation while making high-value patent recognition seem less important. In other words, the proposed method could also be applied to other large-scale data intelligent algorithm problems. The necessity and value of this study are unclear—whether the authors aim to develop a method for high-value patent recognition or merely propose a new intelligent algorithm requires further consideration.
- Based on the main content of the paper, the authors' proposed new method performs better across various metrics. However, the authors should clarify that the core of this study lies in high-value patent recognition. Merely meeting model evaluation metrics does not necessarily prove effective identification of high-value patents—it only means the model passed validation.
- The paper does not provide detailed explanations on whether the training set was labeled for high-value patents, the labeling method used, or the accuracy of the labeling criteria. The quality of the training set significantly impacts the model's application.
- In the conclusion, the paper lacks detailed discussions on the research’s value, significance, and limitations.
- The review of high-value patent recognition methods in the paper merely lists existing approaches without critically pointing out the shortcomings or gaps in current research.
- The core of high-value patent identification lies in discovering truly valuable patents, not merely emphasizing methodological improvements. In empirical research, the ultimate output should be the identified high-value patents themselves, accompanied by an analysis of their value propositions to validate the effectiveness of the proposed method.
Author Response
Comments 1: In the abstract and introduction, the authors only emphasize the importance of high-value patent recognition but lack a description of the research gaps related to methods for identifying high-value patents.
Response 1: Thank you for pointing this out. We agree with this comment. Therefore, we have revised both the abstract and the introduction to explicitly describe the research gaps in existing methods for identifying high-value patents. The changes can be found in the revised manuscript on page 1, paragraph 1, lines 18-22, page 1, paragraph 1, lines 33-36, page 3 , paragraph 2, lines 7-13 and page 3 , paragraph 5, lines 37-41.
Comments 2: In the abstract and introduction, the authors should expand their discussion to clearly explain why EFHO was chosen for high-value patent recognition, including its necessity and effectiveness. Additionally, the authors should address whether other intelligent optimization algorithms could achieve similar results—i.e., the applicability of the proposed method in high-value patent recognition research.
Response 2: Thank you for pointing this out. We agree with this comment. Therefore, we have revised the abstract and introduction to address the necessity, effectiveness, and applicability of EFHO for high-value patent recognition. The changes can be found in the revised manuscript on page 1, paragraph 1, lines 31–33 and page 4, paragraph 1, lines 5–7.
Comments 3: In the abstract and introduction, the description of the research results is overly simplistic. If the proposed new method improves recognition accuracy and effectiveness, the authors should further specify the extent of improvement and its core advantages over other methods to highlight the value of this study.
Response 3: Thank you for pointing this out. We agree with this comment. Therefore, we have revised the abstract and introduction to specify the extent of improvement achieved by EFHO and its core advantages over other methods. The changes can be found in the revised manuscript on page 1, paragraph 1, lines 28–29, page 4, paragraph 1, lines 1–4 and page 4, paragraph 1, lines 8–11.
Comments 4: The authors should provide a systematic review of the applications of the FHO method rather than simply introducing the FHO method itself.
Response 4: Thank you for pointing this out. We agree with this comment. Therefore, we have added a brief systematic review of the applications of the FHO method after the method introduction section. The changes can be found in the revised manuscript on page 6, paragraph 3, lines 4–26.
Comments 5: Provide the full names of PSO, GWO, and WOA.
Response 5: Thank you for pointing this out. We agree with this comment. Therefore, we have We have provided the full names in the revised manuscript (page 10, paragraph 17, lines 41–42).
Comments 6: Table 1 spans multiple pages; it is recommended to adjust the format to display it on a single page.
Response 6: Thank you for the suggestion. We have adjusted the font size and reduced the cell height of Table 1 so that it can now be displayed within a single page when printed or viewed, although it still spans page boundaries in the document layout (page 11).
Comments 7: The authors propose the EFHO algorithm, but the process does not address the problem of high-value patent recognition, focusing more on methodological innovation while making high-value patent recognition seem less important. In other words, the proposed method could also be applied to other large-scale data intelligent algorithm problems. The necessity and value of this study are unclear—whether the authors aim to develop a method for high-value patent recognition or merely propose a new intelligent algorithm requires further consideration.
Response 7: Thank you for pointing this out. We agree that EFHO could be applied to other large-scale data optimization problems, and the primary contribution of this study lies in the algorithmic improvement. High-value patent recognition was chosen as a representative real-world application to validate the method’s effectiveness on structured patent data with multiple indicators. The changes can be found in the revised manuscript on page 19, paragraph 4, lines 24–29 and page 20, paragraph 1, lines 11–16.
Comments 8: Based on the main content of the paper, the authors' proposed new method performs better across various metrics.However, the authors should clarify that the core of this study lies in high-value patent recognition. Merely meeting model evaluation metrics does not necessarily prove effective identification of high-value patents—it only means the model passed validation.
Response 8: Thank you for pointing this out. We agree with this comment. Therefore, we have clarified the positioning of this study to emphasize that high-value patent recognition serves as a representative real-world application for validating the proposed method's effectiveness. The changes can be found in the revised manuscript on page 3, paragraph 5, lines 35–37 and page 19, paragraph 4, lines 24–25.
Comments 9: The paper does not provide detailed explanations on whether the training set was labeled for high-value patents, the labeling method used, or the accuracy of the labeling criteria. The quality of the training set significantly impacts the model's application.
Response 9: Thank you for pointing this out. We agree with this comment. Therefore, we have supplemented the dataset description to explain the labeling process for high-value patents and provided the basis for its accuracy. The changes can be found in the revised manuscript on page 16, paragraph 1, lines 7–12 and page 16, paragraph 2, lines 25–29.
Comments 10: In the conclusion, the paper lacks detailed discussions on the research’s value, significance, and limitations
Response 10: Thank you for the suggestion. We have expanded the conclusion to include a detailed discussion of the research’s value, significance, and limitations. The revisions can be found in the conclusion section of the revised manuscript (page 20, paragraph 1, lines 11–16;page 20, paragraph 2, lines 18–21;page 20, paragraph 3, lines 23–29).
Comments 11: The review of high-value patent recognition methods in the paper merely lists existing approaches without critically pointing out the shortcomings or gaps in current research.
Response 11: Thank you for pointing this out. We agree with this comment. Therefore, we have revised the introduction to explicitly describe the research gaps in existing methods for identifying high-value patents. The changes can be found in the revised manuscript on page 3 , paragraph 2, lines 7-13 and page 3 , paragraph 5, lines 39-41.
Comments 12: The core of high-value patent identification lies in discovering truly valuable patents, not merely emphasizing methodological improvements. In empirical research, the ultimate output should be the identified high-value patents themselves, accompanied by an analysis of their value propositions to validate the effectiveness of the proposed method.
Response 12: Agree. We appreciate the reviewer's emphasis on the practical goal of discovering truly valuable patents. In our study, the positive samples are awarded patents officially recognized by the China National Intellectual Property Administration, ensuring that the evaluation targets are of verified high value (page 16 , paragraph 1, lines 7-10). This definition inherently aligns the model's output with patents of recognized value, eliminating the need for further post-hoc value analysis (page 20 , paragraph 1, lines 11-13).
Reviewer 2 Report
Comments and Suggestions for Authors
- Please clarify more explicitly what the EFHO adds compared to recent variants of the FHO algorithm. What is the precise scientific gap that this work addresses?
- Include a formal statistical analysis (e.g., Wilcoxon test, ANOVA, etc.) of the results in Table 1 to demonstrate whether the observed differences are statistically significant.
- Expand the discussion on how EFHO specifically contributes to the patent classification task. Which variables were most relevant according to the Random Forest model?
- The negative class is defined using a value score of 100 provided by the Patyee database. Could the authors explain how this value is computed? Is this score validated against any external benchmark or standard in the literature? Why was 100 chosen as a cut-off value, and how sensitive are the results to this threshold?
Author Response
Comments 1: Please clarify more explicitly what the EFHO adds compared to recent variants of the FHO algorithm. What is the precise scientific gap that this work addresses?
Response 1: Thank you for pointing this out. We agree with this comment. Therefore, we have clarified the precise scientific gap and the specific contributions of EFHO compared to recent FHO variants in the revised manuscript (page 6, paragraph 6, lines 28–33; page 6, paragraph 7, lines 36–39).
Comments 2: Include a formal statistical analysis (e.g., Wilcoxon test, ANOVA, etc.) of the results in Table 1 to demonstrate whether the observed differences are statistically significant.
Response 2: Agree. We have, accordingly, added a Wilcoxon signed-rank test to evaluate the statistical significance of EFHO's performance differences compared to the baselines. The summarized results and corresponding discussion are presented in the main text (Table 2; page 12, paragraph 2, lines 7–17), and the complete test outcomes are provided in the Appendix.
Comments 3: Expand the discussion on how EFHO specifically contributes to the patent classification task. Which variables were most relevant according to the Random Forest model?
Response 3: Thank you for pointing this out. We agree with this comment. Therefore, additional discussion has been added in the experimental results and conclusion sections to explain EFHO's contribution to the patent classification task (page 18, paragraph 4, lines 30–32; page 19, paragraph 1, lines 1–3;page 20, paragraph 1, lines 5–10).While we did not conduct a separate feature importance analysis, the feature set was determined based on extensive experience in patent analysis and prior research indicating their relevance to patent value.
Comments 4: The negative class is defined using a value score of 100 provided by the Patyee database. Could the authors explain how this value is computed? Is this score validated against any external benchmark or standard in the literature? Why was 100 chosen as a cut-off value, and how sensitive are the results to this threshold?
Response 4: Thank you for your constructive questions. The manuscript has been revised to clarify that the value score is an internal evaluation metric computed by the Patyee database using a proprietary method that is not publicly disclosed. The score of 100 is not used as a classification threshold but rather as a data attribute, the choice of which is mainly driven by two considerations. The changes can be found in the revised manuscript on page 16, paragraph 2, lines 16–25.
Round 2
Reviewer 1 Report
Comments and Suggestions for Authors
The author has made revisions to some of the suggestions, but the following issues still require further consideration:
- The author proposed a method for identifying high-value patents. In addition to demonstrating the effectiveness of the method, I would like to see the actual high-value patents output during the testing of the method, rather than just an evaluation of the model itself. The identified results are not visible.
- The author still has not explained why the proposed model is suitable for identifying high-value patents.
Author Response
Comments 1: The author proposed a method for identifying high-value patents. In addition to demonstrating the effectiveness of the method, I would like to see the actual high-value patents output during the testing of the method, rather than just an evaluation of the model itself. The identified results are not visible.
Response 1: Thank you for pointing this out. We agree with this comment. To make the identified results more visible, we have added a feature importance analysis using the out-of-bag (OOB) permutation method. The new results are presented in Figure 4 and described in the revised manuscript on page 19, paragraph 3, lines 16–27.
Comments 2: The author still has not explained why the proposed model is suitable for identifying high-value patents.
Response 2: Thank you for pointing this out. We agree that it is necessary to clarify why the proposed model is suitable for identifying high-value patents. Accordingly, we have revised the conclusion to explicitly state that high-value patent recognition is essentially a complex classification problem. By leveraging EFHO's strong capability in hyperparameter optimization and Random Forest’s robust classification ability, the EFHO-RF framework provides a suitable and effective solution for this type of task. The changes can be found in the revised manuscript on page 21, paragraph 3, lines 19–23.
Reviewer 2 Report
Comments and Suggestions for Authors
Paper is ready now
Author Response
Thank you very much